# “The Drug Sellers Were Better Organized than the Government”: A Qualitative Study of Participants’ Views of Drug Markets during COVID-19 and Other Big Events

**DOI:** 10.3390/ijerph20021295

**Published:** 2023-01-11

**Authors:** Roberto Abadie

**Affiliations:** School of Global Integrative Studies, University of Nebraska-Lincoln, Lincoln, NE 68588-0368, USA; rabadie2@unl.edu

**Keywords:** COVID-19, Hurricane Maria, big events, natural disasters, PWID, drug markets, harm reduction

## Abstract

“Big events”, such as wars, economic crises, pandemics, or natural disasters, affect the risk environment in which people use drugs. While the impact of big events on injection risk behaviors and access to drug-treatment services is well documented, less is known about the effects of big events on drug markets. Based on self-reporting data on drug availability among people who use drugs (PWUD) in the aftermath of Hurricane Maria in Puerto Rico and during the COVID-19 lockdown in a Midwestern US state, this study aims to document the effects of big events on drug markets. Qualitative data on the effects of Hurricane Maria on drug markets are based on participants’ self-reporting (N = 31). Data collection started after the hurricane and ended in 2020. Data on changes to the drug supply during the COVID-19 lockdown were collected based on semi-structured interviews with PWUD (N = 40) in a Midwestern US state. Findings show that while the drug markets might have initially been affected by big events, most effects were temporary. Drug availability, pricing, and quality might have suffered some initial fluctuations but stabilized as the drug markets absorbed the initial shocks caused by the hurricane and the lockdown measures. In preparation for increasingly more frequent and virulent pandemics and natural disasters, health infrastructures should be strengthened to prevent not only overdose episodes and deaths but also drug-related harms.

## 1. Introduction

In December 2019, the first COVID-19 cases were detected in China, morphing quickly from a local epidemic to a global pandemic. In response, countries adopted containment measures, slowing down economic activities and affecting transportation and commercial hubs. The appearance of COVID-19 on a global scale, along with wars, ecological/natural disasters, and economic crises, constitutes a clear example of what sociologist Sam Friedman terms “Big Events” [1]. Whether they appear suddenly and then wane, for example, hurricanes, or develop and are sustained over time, for example, wars or economic crises, big events can dramatically alter the risk environment [2] in which people who use drugs (*PWUD*), and particularly, people who inject drugs (*PWID*) live, exposing them to an enhanced risk of HIV/HCV transmission, drug overdose episodes and deaths, and related drug-related harms [3,4,5,6,7]. While the impact of big events on PWUD is well documented, less is known about the effects of big events on drug markets. Some have speculated since legal and illegal markets might be linked, it was conceivable that the global slowdown in commercial activity and transportation, which are central to drug markets, could disrupt the circulation of chemical precursors required to produce fentanyl and fentanyl analogs, thereby affecting the drug supply [8]. This assessment seems to be confirmed by a study of PWUD in Georgia that showed that the perceived availability of drugs was reduced during the COVID-19 lockdown, with participants switching to alternative substances if their drug of choice was not available [9]. Other studies also based on self-reporting suggest no changes in drug availability in Germany [10] and Western Balkans [11] or a stable or decreasing drug supply in Australia [12]. With lockdowns limiting mobility and economic activity, some have wondered if drug markets have moved online [13,14] but concluded that while drugs can be bought online with the use of cryptocurrencies and with relative anonymity, drugs are still sourced offline, erecting barriers to the wider use of online markets during COVID-19, particularly among marginalized groups like people who inject drugs (PWID). What little is known about the impact of the pandemic suggests that the drug supply was not severely disrupted—or if so, only temporarily—during the lockdown phase. However, the unprecedented nature of the pandemic, along with a paucity of empirical evidence, renders any preliminary conclusion premature [15].

In September 2017, Hurricane Maria hit the island of Puerto Rico, destroying physical and health infrastructures and disrupting economic activity. In its aftermath, lines at gas stations stretched for miles, and more rural or isolated areas lacked electricity for months. Yet while natural disasters can have devastating effects on the lives of PWID, disrupting social networks and increasing injection risk behaviors and other drug-related harms [16], hurricanes do not seem to have lasting effects on drug availability, pricing, and quality. A study conducted after Hurricane Katrina hit New Orleans showed that according to self-reports the drug supply was briefly interrupted but resumed soon after, as drug dealers moved to other cities along with the displaced population [17,18]. These observations have been confirmed by other studies that showed that drug dealers arriving in new locations after Katrina were able to negotiate market access with already established dealers [19]. A similar market dynamic was documented in the aftermath of Hurricane Sandy in New York City. After a brief interruption, the drug markets recovered relatively quickly [20].

Based on self-reporting data among PWID in the aftermath of Hurricane Maria in Puerto Rico and PWUD and PWID during the COVID-19 lockdown in a Midwestern US state, this study aims to document the effects of big events on drug availability, pricing and quality from the perspective of drug users. Including non-injectors in the study contributes to providing a more complete picture of market demands from the perspective of consumers. While this study focuses on drug demand and not the drug supply side of the drug markets, we believe that there is a close correlation between PWUD’s experiences and the underlying dynamics of drug markets both in the aftermath of Hurricane Maria and the lockdown induced by COVID-19. Understanding changes in the drug supply driven by natural disasters or pandemic events is critical to anticipate changes in patterns of substance use, injection risk behaviors, HIV/HCV risk, and overdose risk and will contribute to the design of evidence-based harm-reduction strategies to mitigate drug-related harms during future big events.

## 2. Materials and Methods

This study employs data from two different sources. Qualitative data on the effects of Hurricane Maria on drug markets are based on participants’ self-reporting (N = 31) and was embedded in a larger study (N = 177) describing the experiences of PWID seeking medication for the opioid-use disorder (MOUD) in rural Puerto Rico in the aftermath of Maria. Data collection started soon after the event, in 2018, and ended in 2020. Self-reporting data on the impact of the COVID-19 lockdown were collected in a Midwestern US state among PWUD in and out of MOUD (N = 40) using a qualitative research design. While most participants in these studies were current PWID some were not.

All participants were at least 18 years old and had either used opioids at least once in the past 30 days or were enrolled in MOUD at the time of the study. Eligibility was determined with the use of a screening questionnaire. Data collection for the Puerto Rican sample was conducted using Respondent Driven Sampling, a proven strategy used to recruit hard-to-reach populations [21,22,23]; the Midwest sample resorted to snowball sampling. The initial recruitment for this study was supported by the Longitudinal Network Cohort (LNC) at the Rural Drug Addiction Research Center of the University of Nebraska-Lincoln.

Interviews were conducted by the author and by trained research assistant with a strong background in mixed methods research and very experienced in conducting research with people who use drugs. In addition to collecting sociodemographic data—age, gender, homelessness, marital status, employment status, and education—we collected data on substance use, such as age at time of the injection, injection frequency, and money spent on drugs. A more detailed sociodemographic background for these populations has been published by Abadie et al. [24]. A semi-structured questionnaire was employed to assess the effects of big events on the drug supply. We collected self-reported data about changes in drug availability, quality, and pricing following Hurricane Maria in Puerto Rico and during the COVID-19 lockdown in a Midwestern US state.

Analysis of the qualitative data was conducted using MAXQDA. Coding was undertaken by the first author and one research assistant working simultaneously and collaboratively. A codebook was used to standardize coding procedures and solve coding disagreements. A priori and emergent codes as well as structural and descriptive codes [25] were developed, taking into consideration the need to account for changes in drug markets in these locations. These codes were iteratively revised and regrouped [26] until they represented a set of higher-level axial codes describing participants’ experiences accessing their drugs of choice in the aftermath of big events. Some examples of these codes are: drug availability; changes in drug availability; pricing; quality; Hurricane Maria; COVID-19 lockdown.

All participants provided informed consent and were compensated for the time and expenses incurred by their participation in the study with amounts ranging from $40 and $70 USD. Approval for data collection in Puerto Rico was obtained by the IRBs of the University of Nebraska-Lincoln and the University of Puerto Rico. The IRB at the University of Nebraska-Lincoln approved the study conducted in the continental United States.

## 3. Results

### 3.1. Sociodemographic Background

Despite the differences between Puerto Rico, a US territory, and a more affluent Midwest US setting, participants in the study share some important sociodemographic characteristics. The Puerto Rican sample has a mean age of 46.7 years and is mostly composed of men (84%), with four out of ten (42%) reporting having experienced homelessness in the past 12 months. In addition, 90% are unemployed or disabled; six out of ten (61%) have not attended college and have only a high school education or less. Slightly more than half (55%) report injecting four or more times a day; approximately one in four (26%) do so between one and three times a day, spending an average of $54 a day acquiring their drugs of choice.

Participants in the Midwest are slightly younger, with a mean age of 40 years, and have a less gendered distribution: 57.5% men and 42.5% women. One-third (35%) were homeless during the past 12 months, and 58.5% were either unemployed or disabled. Almost four out of ten (37.5%) have not attended college, having only completed a high school diploma or less. One in four reports injecting four times or more a day; 27% inject two or more times per week, spending $64 on average per day.

### 3.2. Access to Drugs after Maria

Hurricane Maria devastated the island of Puerto Rico’s infrastructure. Areas were without electricity for weeks and even months, and blocked streets and damaged buildings interrupted access for MOUD and Syringe Exchange Providers (SEP). PWID struggled to cope. However, puntos, as local drug-selling spots are locally known in Puerto Rico, managed to open immediately after the hurricane, despite the fact that illicit drugs are not produced on the island. In preparation for the arrival of the hurricane, some PWID managed to acquire a few extra bags and were able to wait two or three days before returning to the puntos. Waleska [not her real name. All participants in the study are identified by a pseudonym to protect their anonymity and confidentiality] had gathered a “stash”, but when it ran out, she started feeling the painful effects of heroin withdrawal. Anxious, she considered selling her bonsai tree, “the only thing of value that I had”, before her husband managed to find some money. When she arrived at the punto, she saw “a lot of people, traffic jams, desperate people. I saw things that I wish I hadn’t seen. But it was open, el Punto had not closed, it had been open all the time”.

Others, such as Papo, were forced to seek a punto as soon as the hurricane passed: “I had nothing, and I was desperate. I thought: what do I do? I decide to do a run [to the punto] even though I had no money. I jumped through fallen trees and debris, until I saw the dealer, and I tell him, ‘Look, I have no money, I need the bag, I am sick.’ He said, ‘I can’t’, but I insisted, and he finally gave it to me”. Kiri had not prepared for Maria and ran out of his drug of choice almost immediately, but the following day he was able to find what he needed in the local punto. “El Punto never closes”, argues Freddo, explaining why Maria had no effect on drug availability. “There was no food, there was no water, but there were drugs”, noted another participant. In a veiled critique of the slow government response, Vitin, who in his youth had been a local bichote, a bigshot local drug dealer, claimed, “The drug sellers were better organized than the government”.

Not all PWID in our study managed to find their drug of choice right away. Some puntos are located in smaller localities or are disrupted by police or conflicts with rival groups, forcing PWID to travel frequently in search of their drug of choice. Hurricane Maria seems to have exacerbated these issues in some places. Manny believes that the continuous problems in the drug supply in his town, which preceded Maria, are caused by the instability created by rival drug gangs: “After Maria, we went to Caguas and Cidra [other towns] because here there was nothing for months at a time. There it’s the same people running the puntos, the same connection. Here they are destabilized. They are from out of town, from another location, and besides, they are always changing because the guys that were running it before are in jail now. Sometimes you go and it is good, everybody runs there, and then you came back a few days later and it’s not the same! It’s rubbish”.

Miguelito, in another town, struggled to find a dealer because nobody knew where the punto, which had moved because of Maria, would open. “I had to chase them down. I couldn’t find the dealer, everybody was looking for him, and he was in hiding because of the police. It was not easy to buy the first couple of days, and yet the punto never shut down”. Chino, who lives in another location, decided to avoid his local punto and try his luck at a larger venue in a nearby town. Only one day after Maria struck, he found things “all normal, it didn’t lack anything. As if nothing had happened”. Speedball, heroin with a dash of cocaine, is the preferred drug of choice for PWID in rural Puerto Rico, but as it is a blend of two drugs, it often forces study participants to visit more than one punto. Pablito, who like most PWID in rural Puerto Rico uses speedball, struggled to find cocaine in his locality after Maria and was forced to visit numerous venues in nearby towns. “It’s messed up! You have to go here [for cocaine] and there [for heroin]”.

#### 3.2.1. Price

Perhaps reflecting the fact that the drug supply was not significantly altered in the aftermath of Hurricane Maria, drug prices did not suffer any modification. All participants pointed out that in relation to price, things were “the same” and totally “normal”. However, while Maria offered opportunities to PWID to earn money in the informal economy, engaging in debris removal, construction, cleaning, or other activities, particularly in the days that followed Maria’s landfall, some participants struggled to afford their drug of choice. “I use what I can. In a bad day, I use less, what can I do?” (Kiri).

#### 3.2.2. Quality

PWID’s views on the effects of Hurricane Maria on drug quality are shaped by a number of factors, including previous perceptions of drug quality and the simultaneous arrival of fentanyl on the island. Some participants believe that heroin quality had decreased over the years, perhaps nostalgically imagining a mythical past when drug purity was much higher: “Fentanyl is killing people because before there weren’t so many deaths, there were not as many overdoses like today. Before, la droga [heroin] was the size of a tiny match head that cost only $3, got three people high, and you only needed to use only once y te curabas [‘you were cured’, i.e., avoided withdrawal] for that day until the next day. That was quite something! Now, it is chemical stuff” (Miguelito). While fentanyl might have been present on the island before Maria, it was not widely distributed until a few months later. A participant that had been in jail before the arrival of fentanyl remembers his impressions of finding fentanyl in the drug supply for the first time: “It [drug quality] is very different now. Let me tell you something: I never thought that in that year and six months things would change so much. Everything changed because when I went to jail, there was no fentanyl. It’s like kissing the devil because it is much stronger. And it does not last much in the body either; you’ll need to use every fifteen to twenty minutes” (Fray). Study participants initially reacted negatively to the arrival of fentanyl, in part given the risk of overdose death. Some became habituated to its presence, while others sought to avoid it by entering treatment for MOUD or trying to quit “cold turkey”.

Yet, at least in the aftermath of Maria, some participants claimed it was possible to access heroin without fentanyl: “In the *caserio* it is white; it doesn’t have fentanyl. When you put it in the cooker, it does not turn dark like the fentanyl” (Kiri).

### 3.3. Access to Drug Markets during the COVID-19 Lockdown

COVID-19′s effects on drug markets were not uniform. Some locations were more affected than others, and some study participants had difficulties finding their drugs of choice, while others did not. “No, I mean, in this area I don’t think it’s been, COVID hasn’t really been too bad, not like I can imagine it could be in other places” (Johanna). According to methamphetamine users, their drug of choice became harder to find, particularly in more rural or isolated locations: “It made it, made it so much harder to get meth because most people wanted it. So, it seemed like the heroin came, came in, like by the ton then, but meth wasn’t getting through to nobody and everyone was in fucking, like going off the chains, the fucking meth” (Sheila). Even Ralph, with extensive personal connections, had a similar experience: “Oh, yeah. It, it’s really dried up the sources, you know? I mean, it used to be pretty resilient. I mean, you could run around and talk to people, and bam, bam. You had some. Now it’s a major search. I say it’s harder to find.” (Ralph).

#### 3.3.1. Quality

As with availability, participants’ perceptions of the effect of COVID-19 on drug quality are not uniform. Participants’ substance use trajectories and their previous views of drug quality tend to shape their assessments of the impact of the pandemic on the quality of their drug of choice: “The meth has definitely changed from when I first started doing it. When I first started doing it like I got as I told you I was taking the, I’m taking the Vicodins and at first, when I first started doing the meth at 22, it would calm me down and I could concentrate and shit like that but for the last year, I think, maybe two years the quality has changed. I mean, it’s good, but it’s a weird good, like too good. You know what I mean? Like I do just a little bit and be up for two weeks, three weeks at a time and just be absolutely fucking nuts” (Ric).

Frequency of use also affects the perception of drug quality. Participants that use less frequently or that had stopped using for some time before resuming use might be more sensitive to the drug effects given their lowered tolerance: “Yeah, I think it’s gone way down in hell. You know, it’s hard to say because of my age. I feel, so, you know how that U-shape goes up on the using and then your tolerance comes back down. I believe my tolerance has come back down, and I, I think it was better. My, I just think the quality’s better. But it’s pricey” (Daisy).

COVID-19 did not radically affect drug quality. Some changes, like the arrival of fentanyl, pre-existed the pandemic: “I was looking for heroin, but I mainly got fentanyl because it’s cheaper now than actual brown opium. The last thing I used was fentanyl and amphetamine. Amphetamine was cheaper than I’ve ever seen it. The quality of the amphetamine dropped considerably, but the fentanyl stayed fentanyl. I mean there was a little bit more cut in it but quality, not really, I mean it was actually better than the brown stuff I would get, so it’s actually what I ended up looking for. Other than your average amount of cut bags it was about the same” (Sam).

While fentanyl had been present in the heroin supply for some time, the introduction of fentanyl into the methamphetamine markets is more recent. According to Paulie, fentanyl-laced methamphetamine produces a stronger effect, but it wanes faster: “They’re cutting it [methamphetamine] with fentanyl and all kinds of weird shit these days. Fentanyl keeps you high for a minute, up for a minute, and then you just come down, that’s what they do it for so they can make their money faster. It gives you a really good rush, and then you got to go get some more.” (Paulie).

Other methamphetamine users perceive an unreliable drug quality, producing uneven effects: “It sucks, sometimes you do that because you might catch a buzz to some point, sometimes you do the same amount and it gets you the buzz was like too much. I mean you could do the same amount, it’s just the ingredient, I don’t know. You could do the same amount every time but the thing is, it just depends, I had a friend that told me you can go higher, but I do the same amount each time” (Sharon).

Unpredictable effects or more intense cravings due to the introduction of fentanyl into the methamphetamine supply produced some nostalgia for a time when the supply was allegedly purer: “This shit is so badly cut. Back in the day, it was real good. Now it’s just even” (Anderson). Ron found the quality so bad that he stopped using: “The quality went down for sure. It was really expensive and really shitty. So, I quit smokin’ it”.

As some participants recognize, quality during COVID-19 is also affected by access to trusted dealers, something that was also true before the pandemic, as Dora explains: “Some people you don’t want to buy from and some, yeah. You notice real quick who has some good ones and who doesn’t. And who you don’t want to buy from”.

#### 3.3.2. Price

In response to the emergence of the pandemic, economic activity was brought to a halt, initially affecting the drug supply. Yet drug markets seem to have reacted quickly, reconstituting supply and distribution networks. After an initial supply shock, participants were able to find their drugs of choice: “Okay, well see, let me get more specific because at first when COVID started and they shut down the Mexican borders and stuff it got dry as fuck around here and it was impossible almost to find any [methamphetamine]. I was paying $400 dollars for an ounce, I mean it was insane. And so I was using it very sparingly and trying to hold on to it as long as I could, you know? But then it got a little bit easier as things have lightened up, and now I don’t have a problem at all finding it. Before it was worse than pulling teeth, man it was bad” (Sandy).

The initial scarcity drove prices up, at least momentarily: “At first there was kind of a dry drought for a while, and people would go get it places from out of town, so they would charge more. And people pay it. I paid it one time, a little bit more” (Tim). Yet, some prices seem to have stabilized after an initial supply shock. Sheila provides an encompassing description of drug pricing during COVID-19: “[Methamphetamine’s price] is outrageous right now. That’s why heroin is becoming so much more popular. And I’m paying like you know, I was paying 150, 200 an ounce before, and now it’s up to fucking a thousand dollars. Yeah. It was outrageous for a little bit. Heroin got way cheaper and easier to get. I don’t know why that happened, but so did cocaine, too.” (Sheila).

As Paul recognizes, price is not only a factor of supply and demand. What participants pay for their drug of choice is also affected by personal connections to the dealer: “Yeah, it varies, like depending on who I got up for or got it from.” (Paul).

## 4. Discussion

Findings show that while the drug markets might have been initially affected by the disruptions caused by Hurricane Maria in Puerto Rico and the lockdown implemented during the COVID-19 pandemic in the Midwestern United States, most effects were temporary. While drug availability, pricing, and quality might have suffered some initial fluctuations, they tended to stabilize as the drug markets absorbed the shocks caused by the lockdown measures. This finding is consistent with other studies of post-hurricane scenarios in the United States and also, more recently, of the severe lockdown measures adopted in early 2020 by New York City to curb the spread of COVID-19 [27].

Rather than creating an entirely new market dynamic, “big events” like natural disasters or pandemics are superposed to pre-existing local markets. Studies based on self-reporting data about changes in the drug markets in the aftermath of Maria, COVID-19, or other big events should consider that participants’ views are not only rooted in their present experience but are also shaped by their histories and frequency of substance use. In turn, access to participants’ drugs of choice is driven by drug availability, and drug availability and price have shown some correlation although some of the effects might be provisional. One important result of this study is the arrival of fentanyl both in the Midwestern and Puerto Rican settings. Fentanyl seems to be present not only in the context of heroin but increasingly affects polysubstance users of stimulants like cocaine and amphetamine as well. The same trend has been documented elsewhere in the United States [28,29]. This is worrying because fentanyl and fentanyl-related analogs are the main drivers of overdose deaths in the United States, which have been on the rise [30], in part fueled by the presence of COVID-19 [31,32]. Another change in the drug markets during COVID-19 is the emergence of xylazine, a powerful horse anesthetic used to cut opioids that had existed in Puerto Rico for decades [33]. The recent appearance of this powerful drug in the continental United States has been documented in other studies that link it to an increased risk of overdose deaths [34].

Big events like Hurricane Maria and the global lockdown after the emergence of COVID-19 had only a temporary effect on drug markets, showing the limits of the strategy to curb the drug supply that has sustained the War on Drugs for more than fifty years. As other studies have shown, the War on Drugs has had the unintended effect of introducing ever more powerful and dangerous drugs into the market. In particular, the production and distribution of fentanyl and fentanyl-related analogs have recently exploded due to their low cost of production and high potency, yielding high returns for a relatively small volume and partially displacing the heroin trade [35].

While this study is not the first to suggest the need to shift gears in drug policy, moving away from a punitive approach and into one based on harm reduction [36,37,38,39,40,41], with global warming increasing the frequency and potency of hurricanes and a heightened risk of other pandemic events in the future [42,43], these changes are more urgent than ever. In preparation for these events, health infrastructures should be strengthened to prevent not only overdose episodes and deaths but also other drug-related harms. Existing harm-reduction interventions like the implementation of overdose prevention rooms [44,45], ensuring a safe drug supply [46], and less punitive and more accessible (both in terms of cost and availability) MOUD [47] will make significant headway in preparation for the next big event.

Finally, collecting supply-side data in drug markets after big events is notoriously difficult. We suggest that self-reporting data on drug demand after natural disasters or pandemic events can be collected earlier and relatively quickly, offering an important picture of shifting drug markets. These studies might complement epidemiological or criminological data about drug-supply dynamics that are usually produced retrospectively, sometimes months or even years after the event has long passed. This study, based on self-reports of participants’ experiences accessing drug markets during big events has some limitations. While extremely valuable, these data can be affected by recall bias, but more importantly, since drug markets are local in nature, participants’ experiences are shaped by local conditions and, therefore, can be hard to generalize from. In addition, this study focuses on the demand side of the drug markets. While we believe that the perspective of PWUD reflects local market dynamics, other studies are needed to document the effects of big events on the drug supply. Yet despite this limitation, we believe that self-reporting data are critical in mapping sudden changes in drug markets in the aftermath of big events.

## 5. Conclusions

Findings show that while the drug markets might have been initially affected by the disruptions caused by Hurricane Maria in Puerto Rico and the lockdown implemented during COVID-19 in the Midwestern United States, most effects were temporary. While drug availability, pricing, and quality might have suffered some initial fluctuations, they tended to stabilize as the drug markets absorbed the shocks caused by the lockdown measures. In preparation for future pandemics or natural disasters, health infrastructures should be strengthened to prevent not only overdose episodes and deaths but also other drug-related harms.

## Data Availability

All data generated or analyzed during this study are included in this published article.

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
