# Peer review of "“The Drug Sellers Were Better Organized than the Government”: A Qualitative Study of Participants’ Views of Drug Markets during COVID-19 and Other Big Events"

_ijerph, 2023, doi:10.3390/ijerph20021295_

Round 1

Reviewer 1 Report

Thank you very much for the opportunity to review this manuscript.

I must admit that, except for a minor remark, I am very impressed by this text, which I can call a very good article.

In my opinion, this is how good articles should be constructed, which read "in one breath",  whose narrative arouses curiosity and the desire to read "what next?". Besides, a great value of the manuscript is the embedding of the research, in a reality familiar to all of us. In addition, for me, as a sociologist and addiction specialist, the text is highly cognitive and stimulates the design of analogous studies in my country. So, I also thank you for the inspiration and hope that other readers will have similar impressions and thoughts.

There is one thing that you don't suit me very well. In my opinion, if one is describing results collected from such small samples, it seems a bit unnatural to give data in %. As a sociologist, I was taught that then we give frequencies, not percentages. (in  3.1. Sociodemographic background)

I think it will look more reliable.

Author Response

Many thanks for your comments. We have chosen to provide results in % because this is the preferred modality for reporting results in this journal. 

Reviewer 2 Report

This paper was interesting and timely - there is much merit to studying the robustness of drug markets/drug availability and drug-use practices in post-crises situations. The methodology is sound and the findings are relevant and important. Where I think the paper does need more work is in its focus and contextualisation.

The authors seem unsure if they are talking about PWUD or PWID! It seems that all of the Hurricane sample and most (if not all) of the Covid sample are PWID (or people who have injected drugs) rather than 'mere' PWUD. This is an important group to study - arguably the risks associated with disrupted drug markets are greater for those WID than those WUD as (a) 'addiction' means this group are more likely to desperately seek drugs, and (b) the health risks related with injecting behaviours are greater (the paper mentions reports of fentanyl and xylazine becoming more prevalent in the circumstances discussed, with obvious risks relating to OD and/or substance substitution). If the paper can be framed as focusing on PWID then it should.

Related, the (currently very brief - see below) review of drug-market changes after Covid/disasters should be more nuanced, reflecting how different changes are both more likely and more likely to be policy/risk relevant for 'harder' drug markets (such as for PWID) than for 'softer'/recreational drug markets. The move to cryptomarkets, for example, is much less likely/much less significant for PWID than PWUD (esp. PWUD recreationally) for at least two reasons. PWID drugs are more likely to want their drugs quickly (to alleviate 'addiction') and therefore prefer FtF markets than (delayed) delivery from cryptomarkets. Secondly, PWID are more likely to be marginalised in other ways (some of which are documented in the samples descriped here) - i.e., less likely to have the economic/social/cultural capital needed to access drugs online. A final point here is that there is definitely more literature out there looking at how drug markets (and/or drug use) have (or have not) been disrupted by Covid than is currently mentioned. See, e.g., https://www.emcdda.europa.eu/topics/covid-19_en - but note that this includes only some examples!

Author Response

Many thanks for your comments and suggestions. This paper presents data from two different studies. A study of PWID in rural Puerto Rico in the aftermath of Hurricane Maria, conducted between 2018 and 2020, and a more recent study on PWUD in rural Nebraska during the Covid-19 lockdown. While most of the PWUD in rural Nebraska included in the study are PWID, not all are. We believe that since the study is based on participants' perceptions and drug-seeking behaviors in the aftermath of big events, including some participants that are non-injectors brings perspectives about drug markets that extend the experiences of PWID, providing a more robust set of findings. We have added this rationale at the end of the introduction section.

Thanks for noticing that changes in drug markets are more likely to affect PWID than others, particularly the reliance on crypto markets. We have now amended the introduction to incorporate this observation.

We are also thankful for the suggestions to update the literature on the effects of Covid-19 on drug markets. We have now included additional references.

Reviewer 3 Report

Thank you for the opportunity to review this interesting paper. I can see it providing a contribution to the literature. A few minor suggestions:

- Please capitalize PWUD and PWID when you include it the first time. PWID is not explained in the paper despite it being a new acronym.

- In the introduction, please clarify the effects these big events have on formal AND informal drug markets, as they are often related

- In the introduction, please clarify most marginalized persons cannot purchase drugs online. 

- More information on methods is needed. Who did the interviews? How were participants recruited? How were participants "compensated" for their time? Was the ultimate research question relating to these big events or did this data come out as part of a larger study? Please provide a few examples of the codes used during coding.

- Why have you blinded the midwestern US city? Are you choosing not to provide more information on "rural" PR for the same reason? Explain.

- Are all participant names pseudonyms? Clarify

- When talking about the drug mix for speedballs, does this mean participants mix the drugs themselves?

- Was the drug supply in PR less affected because drugs may be produced there as opposed to being primarily illegally transported into the continental US? Some additional context would be beneficial for less familiar readers

- It appears the data under "price" for the USA show spikes because of limited access or a dried-up market. This is not as clearly relayed in the supply section. Are there better data you can use to illustrate these points? Indeed, price and availability are intimately connected. 

Author Response

Many thanks for your comments and suggestions.

We have revised the manuscript, capitalizing PWID and PWUD the first time we introduced these terms.

Thanks for suggesting that sometimes legal and illegal drug markets are connected. We have amended the manuscript to describe the effects of Covid-19 on producing the chemical precursors required to produce fentanyl. While some hypothesized that the economic disruptions might affect the production and distribution of fentanyl, evidence suggest that, at least for the U.S., this is not the case.

Thanks for pointing out that marginalized PWID and other groups are unlikely to access online markets. We have now made this point clear in our revision.

Regarding the methods, we have revised the section to explain that interviews were conducted by trained research assistants and that payment had been provided to study participants. We have clarified that data about participants' perceptions of drug markets during big events were collected as part of more extensive studies. Data from drug markets in Puerto Rico was collected as part of a study on social networks, intravenous drug use, and HIV/HCV risk conducted between 2015 and 2019. Data among PWUD in a midwest location was collected within a larger study on the effects of Covid-19 on access and retention to Methadone and Buprenorphine programs conducted in 2021.

Following your suggestion about including examples of some of the codes produced during the analysis, we have amended the methods section to include some of our significant codes.

All participants names are pseudonyms to protect participants' confidentiality and anonymity. For the same reason, we have chosen not to name the locations where the studies were conducted.

We have clarified that participants themselves mix the drugs to prepare the speedball. Of course, when drugs are bought, they might come with other impurities or compounds not chosen by the participants.

Illegal drugs are introduced to Puerto Rico from the neighboring Dominican Republic via Mexico and sometimes shipped to the continental U.S. No drugs are produced in Puerto Rico. We have added a paragraph providing more context to clarify this issue.

Finally, we have clarified that price and availability are related.

Round 2

Reviewer 2 Report

One further (very minor - apologies for not picking it up on the first review) change to make: The first Covid-19 cases were detected in December 2019, not January 2020 (hence being called Covid-19...)!

Author Response

Many thanks for picking this up. The manuscript has been amended to indicate the correct date in which the first cases of Covid-19 were detected.